# Rheological Properties of SBS/CRP Composite Modified Asphalt under Different Aging Treatments

**DOI:** 10.3390/ma13214921

**Published:** 2020-11-02

**Authors:** Shuhua Wu, Rui He, Huaxin Chen, Wenke Li, Guanghui Li

**Affiliations:** 1School of Civil Engineering and Architecture, Zhengzhou University of Aeronautics, Zhengzhou 450046, China; wsh@zua.edu.cn (S.W.); lgh@zua.edu.cn (G.L.); 2School of Materials Science and Engineering, Chang’an University, Xi’an 710061, China; hxchen@chd.edu.cn; 3Henan Province Highway Engineering Bureau Group Co., Ltd., Zhengzhou 450052, China; liwenke2000@163.com

**Keywords:** road engineering, crumb rubber powder, SBS/CRP composite modified asphalt, aging treatment, rheological properties

## Abstract

The objective of this study is to evaluate the rheological properties of SBS/CRP composite modified asphalt under different aging conditions in alpine cold regions. In this study, the styrene-butadiene-styrene (SBS) block copolymer, crumb rubber powder (CRP), softeners and various additives were used to prepare the SBS/CRP composite modified asphalt. The 4.5% SBS modified asphalt and SK90 virgin asphalt were also selected for comparing. The thin film oven test (TFOT) and pressure aging vessel (PAV) aging test were conducted to simulate the different aging conditions. The frequency sensitivity, high-temperature deformation resistance and low-temperature crack resistance of SBS/CRP composite modified asphalt under different aging conditions were studied by the dynamic shear rheometer (DSR) and bending beam rheometer (BBR) test. The results show that the frequency sensitivity of SBS/CRP composite modified asphalt is the lowest. At low and medium temperatures, it has relatively low strength and good flexibility. The master curve of composite modulus for the unaged and aged SBS/CRP composite modified asphalt is in a cluster state. It has strong anti-aging capability, which is less affected by aging conditions. It also has a strong low-temperature relaxation ability, which can meet the requirements of performance at −24 °C in PG classification. The high-temperature deformation resistance of SBS/CRP composite modified asphalt is not much different from SBS modified asphalt’s. Therefore, it can provide the basic support for the application in alpine cold regions under the conditions of low temperature, strong radiation and heavy load.

## 1. Introduction

The styrene-butadiene-styrene (SBS) modified asphalt with good high-temperature stability, low-temperature flexibility, aging resistance and fatigue resistance is widely used in road engineering [1,2,3]. However, it has an unsaturated C=C bond so that its low-temperature performance and aging resistance are slightly insufficient in special regions, such as alpine cold plateau regions with strong ultraviolet radiation [4,5]. Meanwhile, the demand for SBS modifier is increasing and leads to the increase in price with the development of highway construction [6]. In addition, it is difficult to recycle the SBS polymer because of the preparation process of modified asphalt, which is not conducive to the sustainable development. At the same time, the crumb rubber powder (CRP) was used more and more popular in the field of road engineering. Some studies concluded that when the rubber powder is mixed with virgin asphalt, cracks appear at the interface, which passivates and consumes energy. The flexibility of virgin asphalt is thus improved. Moreover, elasticity restoration, anti-aging, oxidation resistance and many other advantages have been improved for the original asphalt [7,8,9,10]. It is widely acknowledged that CRP modified asphalt can improve the high-temperature performance, rutting resistance, and aging resistance of asphalt pavement [11]. Rasool et al. [12] analyzed the aging and rheological properties of waste rubber modified asphalt. The results showed that the rubber powder can further enhance the low-temperature performance and anti-aging properties of SBS modified asphalt. Therefore, it is an effective method to combine the crumb rubber powder (CRP) with the SBS as bi-modifier to improve the performance of the SBS modified asphalt binder, reduce the content of SBS in the modified asphalt and promote the recycling of resources.

The aging phenomenon is already a complex process in neat asphalt, and its complexity increases when composite modification is involved. A large number of researchers used laboratory accelerated aging methods such as thin film oven test (TFOT) and pressure aging vessel (PAV) test to simulate the short-term aging and long-term aging, respectively [13,14,15,16,17]. Conventional experiments including the penetration test, softening point test, ductility test, Fraass breaking point test and rheological methods including the dynamic shear rheometer (DSR) test, bending beam rheometer (BBR) test and rotational viscosity (RV) test were widely used to evaluate the paving performance and rheological properties of asphalt binders [18,19,20,21,22]. In cold regions, asphalt pavement is more prone to cracking because of the aged asphalt binders show the characteristics of brittleness and hardening [23]. Hence, it is of great importance to have a better understanding of changes in rheological properties and cracking resistance of composite modified asphalt binders after aging. Some studies on the composite modified asphalt binders were conducted. Guo et al. [24] studied and concluded the best preparation technology of SBS/rubber powder modified asphalt. Li et al. [25] studied the aging resistance of SBS/CRP-modified asphalt and found that the aging resistance of SBS/CRP-modified asphalt was inferior to that of rubber powder modified asphalt. Tan et al. [26] used seven kinds of asphalt binders and different modifiers to study the viscoelastic influences on the basis of the hysteresis loop analysis; they found that the rubber powder (or SBS) and aromatic oil significantly improved the elastic energy storage of asphalt. Wang et al. [27] used the SBS and rubber powder to prepare composite modified asphalt. The swelling temperature for two mixed modifiers was 180 °C, and the swelling time was 1 h; the shearing temperature was 180 °C, and the shearing time was 1 h; the breeding temperature was 170 °C, and the breeding time was 3 h. They also concluded that adding some inorganic or organic powders can play a role of anti-aging, such as rubber powder, nano-TiO_2_ and carbon black, etc. However, the previous researchers tend to focus on parts of the performances of the composite modified asphalt binder and are lack of the system evaluation of the rheological properties of SBS/CRP composite modified asphalt from the aspects of high-temperature and low-temperature performances under different aging treatments. Meanwhile, the conditions of these modified asphalts in alpine cold regions are rarely considered [28,29,30].

The objectives of this study are to improve low-temperature cracks and anti-aging ability of asphalt used in alpine cold regions; therefore, SBS and CRP were used to modify the asphalt in this study. Softeners were added to reduce the viscosity of modified asphalt, and various additives were also added to improve the stability and aging resistance of asphalt. In this study, the virgin asphalt, SBS modified asphalt and SBS/CRP-modified asphalt were all treated by the thin film oven test (TFOT) and pressure aging vessel (PAV) aging. Based on the time-temperature equivalent characteristic, frequency and temperature sensitivity were analyzed. The anti-aging, high and low-temperature rheological properties of virgin asphalt, SBS modified asphalt and SBS/CRP modified asphalt were also evaluated.

## 2. Raw Materials and Test Methods

### 2.1. Raw Materials

#### 2.1.1. Virgin Asphalt

The SK90 asphalt, which produced SK Group from Seoul, Korea, was used as the virgin asphalt, and technical parameters of its basic performances are shown in Table 1.

#### 2.1.2. SBS Modifier

The YH791was selected as the SBS modifier, which produced in Sinopec Baling Petrochemical Co., Ltd. (Yueyang, China). The basic performances are shown in Table 2. The content used for preparing SBS modified asphalt is 4.5% based on the previous study [31].

#### 2.1.3. Crumb Rubber Powder

The crumb rubber powder was mesh 40–60, and purchased from Changsheng Rubber Factory, Henan, China. Its various indexes are shown in Table 3. The physical indexes and chemical indexes were tested based on the Chinese specification [32].

#### 2.1.4. Solubilizer, Anti-Aging Agent and Auxiliary Agent

The solubilizer is an oil-rich in saturated and aromatic components, which produced in Shandong, China. It can make the crumb rubber powder and SBS in asphalt fully swell and reduce the viscosity of asphalt. The contents of solubilizer, anti-aging agent (BC-1 anti rutting agent, which produced in Shandong, China) and auxiliary agent (Sulfur, which produced in Zhengzhou, China) were 7%, 3% and 0.2% by mass of virgin asphalt, respectively [31].

#### 2.1.5. SBS/CRP Composite Modified Asphalt

The preparation process of SBS/CRP modified asphalt referenced to the literature [31]. The virgin asphalt was firstly heated and kept at 180 °C. The rubber and anti-aging agent were added into the virgin asphalt and mixed by the shearing machine (The type was A30 and the shearing rate was 5000–25,000 rpm, which produced in Shanghai Ouhe Machinery Equipment Co., Ltd., Shanghai, China) for 30 min; the auxiliary was then added and mixed for 30 min. After that, the SBS was added; they were blended for 25 min and then sheared for 30 min.

### 2.2. Preparation of Aged Asphalt

The aging process included short-term aging and long-term aging. The short-term aging was conducted by the TFOT to simulate the aging of asphalt in the storage, transportation, mixing and compaction during the construction. The conditions of TFOT for the specimen were 5 h at 163 °C according to the requirements of specification ASTM D-1754 [37]. The long-term aging was conducted by PAV test to simulate the aging of long-term service performance of asphalt. The conditions of PAV test for the specimen were 20 h at 100 °C, 2.1 MPa based on the ASTM D-6521 [38].

### 2.3. DSR Test

In order to evaluate the time-temperature equivalent characteristic of asphalt binder before and after aging, the frequency sweep test was carried out by the dynamic shear rheometer (DSR) (The type was MCR 302 & MCR 102, which produced in Anton Paar company, Ostfildern, Germany) [39,40,41]. The frequency range was 0.1–100rad/s, and the temperature range was −20~70 °C.

### 2.4. BBR Test

The bending beam rheometer (BBR) (The type was TE-BBR, Canon company, Melville, NY, USA) test was used to evaluate the low-temperature performance of SK90, SBS and SBS/CRP modified asphalt before and after aging. The test temperatures were −12, −18, −24, and −30 °C, respectively. Three parallel tests were used for each condition.

Based on the loading time and original data of deformation of asphalt before and after aging, the relationship (1) between the creep compliance and time was obtained to further characterize the low-temperature rheological properties of modified asphalt. It was fitted based on the Burgers model to obtain the viscoelastic parameters. The comprehensive compliance parameter *J* was used to evaluate the low-temperature relaxation characteristics of asphalt binder from the material itself.
(1)$$S(t)=\frac{Pl^{3}}{4bh^{3}v(t)}=\frac{l}{D(t)}$$
where: *v*(*t*) is the deformation in the middle of the beam, mm; *b* is the width of the beam, 12.70 mm ± 0.05 mm; *P* is the beam’s fixed load, 980 mN ± 50 mN; *h* is the height of beam, 6.35 mm ± 0.05 mm; *l* is the span of beam, 102 mm.

Burgers model can be expressed as the relationship (2).
(2)$$J(t)=\frac{1}{E_{1}}+\frac{1}{E_{2}}(1-e^{\frac{-tE_{2}}{\eta_{2}}})+\frac{t}{\eta_{1}}$$
where: $E_{1},E_{2},\eta_{1},\eta_{2}$ are the parameters of Burgers model.

The Burgers creep equation can be described as the relationship (3):(3)$$J(t)=J_{E}+J_{De}^{}+J_{V}$$
where:$J_{V}$ is creep compliance of asphalt binder; *J_E_* is transient or glassy shear compliance; $J_{De}$ is delay compliance.$J_{E}=\frac{1}{E_{1}}$,$J_{De}=\frac{1}{E_{2}}(1-e^{\frac{-tE_{2}}{\eta_{2}}})$, $J_{V}=\frac{t}{\eta_{1}}$.

The comprehensive compliance parameter $J=J_{V}(1-\frac{J_{E}+J_{De}}{J_{E}+J_{De}+J_{V}})$ can characterize the low-temperature performance of asphalt materials.

### 2.5. MSCR Test

The multiple-stress creep-recovery (MSCR) test was carried out to quantitatively analyze the high-temperature deformation resistance of three asphalts. Two shear stresses, 0.1 and 3.2 kPa, were used, respectively. It has 10 cycles, and each cycle includes a creep stage of loading for 1 s and a recovery stage of unloading for 9 s. The total time for the entire test was 300 s, and the first 100 s was used to condition the specimen; the test temperature is 64 °C.

The single-cycle non-recoverable creep compliance *J_nr_* was calculated based on the collected strains, which can can be expressed as the relationship (4). The change rate of *J_nr_* with stress *J_nr-diff_* and the average strain recovery rate *R* were used to evaluate the high-temperature performance of asphalt binder. The calculation method is as follows the relationship (5) and (6):(4)$$J_{nr}=\frac{\varepsilon_{r}-\varepsilon_{0}}{\tau }$$
(5)$$J_{nr-diff}=\frac{J_{nr(3.2)}-J_{nr(0.1)}}{J_{nr(0.1)}}$$
(6)$$R=\frac{\gamma_{r}}{\gamma_{p}}$$
where: *ε*_0_—initial deformation; *ε_r_*—remaining deformation after 9 s recovery; *τ*—shear stress; *J_nr_*—single non-recoverable creep compliance; *J_nr-diff_*—change rate of non-recoverable creep compliance with stress; *R*—recovery rate of single strain; *γ_r_*—deformation of recoverable part (including delayed elastic deformation and elastic deformation); *γ_p_*—deformation of loading for 1 s.

*J_nr_*_(0.1)_ and *J_nr_*_(3.2)_ were used to represent the average value of non-recoverable creep compliance in 10 cycles under the stresses of 0.1 and 3.2 kPa, respectively. They could reflect the permanent deformation resistance of asphalt under different stresses. The smaller the value, the better the high-temperature performance of asphalt. *R*_(0.1)_ and *R*_(3.2)_ represent the average value of deformation recovery rate within 10 cycles under the stresses of 0.1 and 3.2 kPa, respectively.

Based on the above description, the whole test schematic of this study is shown in Figure 1.

## 3. Results and Discussion

### 3.1. Analysis of Time-Temperature Equivalent Characteristic

#### 3.1.1. Change of Composite Shear Modulus

Double logarithmic coordinates were used in the frequency domain. The complex shear modulus of nine specimens at different temperatures was plotted, as shown in Figure 2. It is found that the sensitivity of composite shear modulus of each specimen varies greatly under different temperature intervals at the fixed frequency. At −20 and −10 °C, the composite shear modulus of the same specimen is very close, and two curves almost overlap; at 10–70 °C, the composite shear modulus changes more sensitive; at the medium temperature region of 10–30 °C, the change is the most sensitive. At different frequencies, the slope of the curve represents the sensitivity of material to loading frequency. The slope of the curve at a low temperature is almost zero. The higher the temperature, the greater the slope of the curve. It is very sensitive to the loading frequency. The asphalt materials in the middle temperature regions are more sensitive to the temperature than the frequency.

In Figure 2a–c, it can be seen that the complex modulus of original sample SK90 ranges from 5 Pa to 433 MPa. After short-term aging, it changes from 10 Pa to 402 MPa. After long-term aging, its range is from 82 Pa to 433 MPa. The complex modulus increase in short-term and long-term aged SK90 is 69.7%, 1349.7% compared to the original SK90 at 0.1 rad/s. The virgin asphalt has no significant influence on the low-temperature region. After long-term aging, the asphalt tends to become hard, which has a significant influence on the low-frequency and high-temperature region.

As shown in Figure 2d–f, the complex modulus of original SBS modified asphalt is from 122.4 Pa to 298 MPa.After short-term aging, it changes from 203.4 Pa to 348 MPa. After long-term aging, it ranges from 135 Pa to 456 MPa. The complex modulus increase in short-term and long-term aged SBS modified asphalt is 66.2%, 10.3% compared to the original SBS modified asphalt at 0.1 rad/s. It increases significantly in the low-frequency and high-temperature regions after short-term aging, but decreases after long-term aging. It increases significantly in the high-frequency and low-temperature regions. The main reason is that the butadiene soft segment of SBS modifier of SBS polymer breaks after short-term aging, and its hard segment is dispersed in the asphalt, which increases the elasticity of SBS modified asphalt. After long-term aging, the SBS modifier degrades, and its temperature sensitivity is similar to that of virgin asphalt. The composite shear modulus decreases at the conditions of high temperature and low frequency.

From Figure 2g–i, it is shown that the complex modulus of original SBS/CRP modified asphalt ranges from 399.4 Pa to 277 MPa. After short-term aging, it changes from 176.8 Pa to 326 MPa. After long-term aging, it is from 181 Pa to 290 MPa. The complex modulus increase in short-term and long-term aged SBS/CRP modified asphalt is −55.7%, −54.7% compared to the original SBS/CRP modified asphalt at 0.1 rad/s. After short-term aging, the SBS/CRP composite modified asphalt has certain sensitivity in the low-frequency region. The main reason is that the crumb rubber powder further swells and decomposes under the action of thermo-oxidation in short-term aging, and unsaturated double bonds break. The composite shear modulus decreases in the low-frequency region, and changes slightly in the high-frequency region. It is indicated that the complex moduli in the low-frequency region after long-term aging and short-term aging change less, and it reflects the anti-aging ability of SBS/CRP composite modified asphalt. In the high-frequency region, the complex modulus of SBS/CRP composite modified asphalt is 63% that of SBS modified asphalt. It is indicated that the SBS/CRP composite modified asphalt has lower stiffness and stronger deformability than the SBS modified asphalt at a low temperature, which is not easy to crack. This may be caused by the forming of the three-dimensional network structure in the asphalt added with SBS/CRP.

#### 3.1.2. Analysis of Master Curve of Composite Shear Modulus

Based on the inverse curve function model of time-temperature equivalent principle, the temperature −10 °Cwas taken as the reference temperature, and the shift factor at other temperatures was calculated, as shown in Table 4. Since the asphalt was in a non-linear viscoelastic range at 70 °C, 70 °C was not considered in generate the master curve of composite shear modulus.

The measured composite shear modulus at −20, 10, 30, and 50 °C were translated, respectively, and the master curve was obtained. In order to compare and analyze the effect of aging on the performance of asphalt, the master curve of composite shear modulus before and after aging was drawn, as shown in Figure 3a–c.

In Figure 3a, the master curve of composite shear modulus of SK90 before and after aging changes a lot in the low-frequency region, which tends to cluster gradually in the high-frequency region. The effect of SK90 on the composite shear modulus in the high-frequency and low-temperature region is weakened. In Figure 3b the effect of SBS modified asphalt on the composite shear modulus is smaller, and the curve separation occurs only in the low-frequency region. In contrast, Figure 3c, three curves of SBS/CRP composite modified asphalt are the most stable during the change process of frequency. They are almost in the state of aggregation in the entire frequency range, and slightly affected by the aging. Therefore, the SBS/CRP composite modified asphalt has the best anti-aging performance.

### 3.2. Analysis of Low-Temperature Creep Characteristic Based on the Burgers Model

The BBR test results of each specimen at different temperatures are shown in Table 5. It can be seen that, as the degree of aging deepens at the same temperature, the stiffness modulus S of asphalt increases and the creep rate m decreases. Test results of SK90 virgin asphalt, SBS modified asphalt and SBS/CRP composite modified asphalt could meet the requirements of the Strategic Highway Research Program (SHRP) specification at −12, −18 and −24 °C, respectively. When the test temperature is −30 °C, the movement of asphalt segment is limited by freezing. When the asphalt is hardened, the value S of specimens in all groups is more than 300 MPa, and the value m is less than 0.3.

The viscoelastic parameters of Burgers are shown in Table 6. The comprehensive compliance parameters were calculated from the data of Table 6, as shown in Table 7. Under a low-temperature condition, the proportion of elastic deformation of asphalt is larger, and the proportion of its viscous deformation is reduced. At this time, if the proportion of viscous flow of asphalt is higher, the shrinkage tensile stress of material can be relaxed by the flow method, thus reducing the low-temperature shrinkage of asphalt pavement. Therefore, in a low-temperature environment, the larger the value *J* of asphalt, the better the low-temperature crack resistance of asphalt. It can be seen from Table 6 that the value *J* of SBS/CRP composite modified asphalt is 1.04 and 1.55 times that of SBS modified asphalt and SK90 virgin asphalt, respectively. It is indicated that the composite modified asphalt has stronger low-temperature relaxation ability and crack resistance than the SBS modified asphalt and SK90 virgin asphalt, which is consistent with the results presented by Zhou et al. [42].

### 3.3. Analysis of MSCR Test Results

#### 3.3.1. MSCR Test Curve

MSCR test curves of SK90 asphalt, SBS modified asphalt, and SBS/CRP composite modified asphalt with no aging, TFOT aging, and PAV aging are shown in Figure 4a–c. It is found that as the aging degree deepens, the cumulative strain gradually decreases. The changing amplitude of cumulative strain at 0.1 kPa is larger than that of cumulative strain at 3.2 kPa. The cumulative strain gaps of three asphalts after long-term aging are further narrowed.

#### 3.3.2. Analysis of Values *J_nr_* and *R*

The *J_nr_*_(0.1)_, *J_nr_*_(3.2)_ and *R* of three asphalts at different aging conditions are shown in Figure 5.

It can be drawn from Figure 5a–c that, under the same stress, the *J_nr_* of SBS modified asphalt is larger than that of SBS/CRP modified asphalt, but the *R* changes in the opposite trend. The larger the unrecoverable creep compliance *J_nr_*, the smaller the strain recovery rate *R*. As the stress increases to 3.2 kPa, the *J_nr_* increases at different degrees, and the *R* continuously decreases, indicating the effect of heavy load on the permanent deformation resistance of asphalt. For modified asphalts, whatever the aging conditions are, their outstanding advantages are higher elastic recovery rate and less unrecoverable creep compliance. It is shown that the modified asphalt has stronger rutting resistance. In addition, the elastic recovery rate of SBS/CRP modified asphalt is better than that of SBS modified asphalt, and its unrecoverable creep compliance *R* is smaller. However, after TFOT aging, *J_nr_* and *R* of these two modified asphalts are relatively smaller. It is indicated that the high-temperature performance of SBS/CRP modified asphalt and SBS modified asphalt is not much different.

However, the *R* of virgin asphalt is abnormal. The strain of original virgin asphalt does not recover or recovers a little, and continues to increase with the unloading of the two stresses for 9 s. For the original virgin asphalt and specimens after short-term aging, when the stress is 3.2 kPa, the strain continuously or step wisely increases. For example, the original asphalt SK90 only slightly recoveries at the first five cycles under 0.1 kPa for 20 cycles at two stress levels. 

For the virgin asphalt and part of specimens after TFOT, the strain does not recover under most cycles after unloading, especially under the shear stress of 3.2 kPa. The main reason may be that the virgin asphalt itself is a viscoelastic material. The viscosity of original virgin asphalt itself at 64 °C is very small. Under the action of load and temperature, the force between asphalt molecules is very weak. When large shear stress is unloaded, the strain of asphalt continues to increase after 9 s due to the large inertial force. The time and amplitude are also different. After short-term aging and long-term aging, the viscosity of asphalt increases. When large shear stress is unloaded, the inertia of asphalt reduced due to the increase of internal deformation resistance of molecules. The strain increases slightly, and gradually recovers partly, which reflects the increase in strain recovery rate. It is also possible that when the stress increases to 3.2 kPa, the virgin asphalt is likely in a non-linear range, which is not suitable for such test conditions. Therefore, the relative difference *J_nr-diff_* of unrecoverable creep complianceof asphalt with the stress is not comparatively analyzed.

#### 3.3.3. *J_nr-diff_* Analysis

The relative difference *J_nr-diff_* of unrecoverable creep compliance was used to further analyze the stress sensitivity of SBS and SBS/CRP modified asphalts. The larger the value *J_nr-diff_*, the higher the stress sensitivity of asphalt. It can be seen from Figure 6 that the *J_nr-diff_* of SBS modified asphalt increases first and then decreases as the aging degree increases. However, the *J_nr-diff_* of SBS/CRP composite modified asphalt slightly increases after short-term aging and significantly decreases after long-term aging. The *J_nr-diff_* of virgin asphalt and SBS/CRP modified asphalt after PAV aging are higher than that of SBS modified asphalt. The stress sensitivity of SBS modified asphalt after short-term aging is slightly higher than that of SBS/CRP modified asphalt. It is indicated that the SBS/CRP modified asphalt has more prominent stress sensitivity of permanent deformation resistance under high-temperature conditions, but the *J_nr-diff_* of these two asphalts under different aging methods is slightly different.

By analyzing these three indexes *J_nr_*, *R* and *J_nr-diff_*, it can be concluded that the high-temperature deformation resistance of both SBS/CRP composite modified asphalt and SBS modified asphalt is not much different, which is consistent with the results presented by Li et al. [25]. It can meet the requirements of high-temperature performance in alpine cold regions.

## 4. Conclusions

In this study, SBS/CRP-modified asphalt was prepared. Rheological Properties, such as the high-temperature performance and low-temperature performance of original asphalt, SBS-modified asphalt and SBS/CRP-modified asphalt under different aging conditions were evaluated and analyzed. The following conclusions are drawn:(1)Short-term aging has no significant influence on the SBS/CRP modified asphalt. The SBS/CRP asphalt has lower strength and stronger flexibility at low and medium temperature. Its composite shear modulus changes less with the temperature. It has good temperature sensitivity.(2)The stiffness modulus and creep rate of SBS/CRP composite asphalt at −24 °C can meet the requirements of specification, which improves the low-temperature performance of asphalt binder. Based on Burgers rheological model, the comprehensive creep compliance parameter *J* can better evaluate the low-temperature performance of asphalt.(3)The stress sensitivity of SBS modified asphalt after short-term aging is slightly higher than that of SBS/CRP composite modified asphalt. However, the stress sensitivity of original asphalt and SBS asphalt after PAV aging is lower than that of SBS/CRP composite modified asphalt. It is indicated that the stress sensitivity of SBS/CRP modified asphalt under high-temperature conditions is more prominent. The repeated creep performance of two asphalts under different aging conditions is slightly different. It can meet the requirements of asphalt binder of high-temperature performance under heavy load conditions in alpine cold regions.(4)Considering economic factors, the SBS/CRP composite modified asphalt can meet the requirements of low-temperature performance at −24 °C, which is benefit to most of alpine cold regions. A special climate environment in alpine cold regions should be considered during the construction, and the composite modified asphalt has higher viscosity.

## Figures and Tables

**Figure 1 materials-13-04921-f001:**
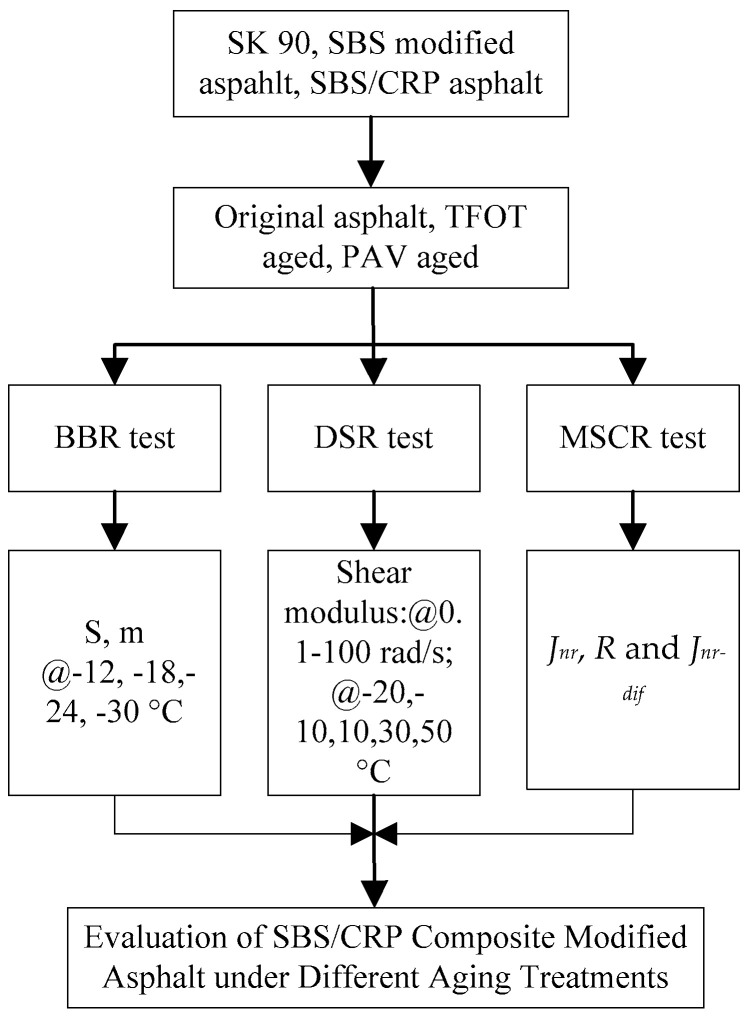
Test schematic of this study.

**Figure 2 materials-13-04921-f002:**
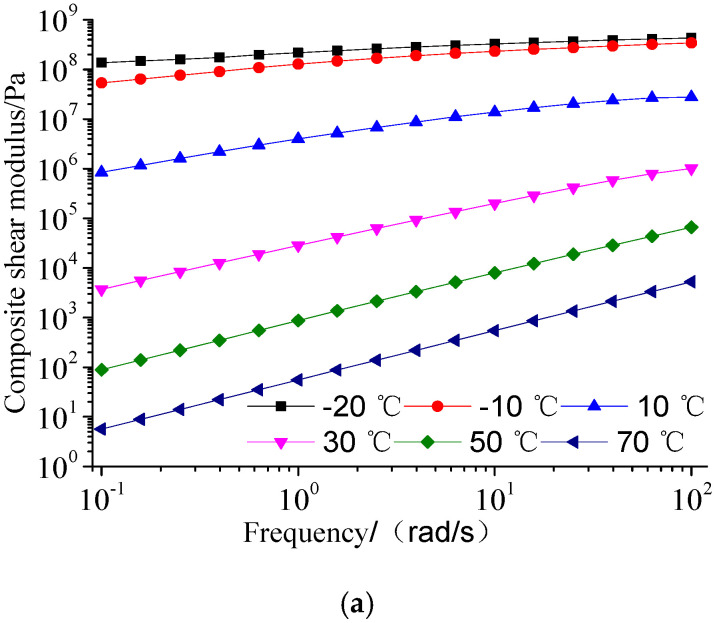
Composite shear modulus of three asphalts with frequency before and after aging: (**a**) SK90 (unaged specimens); (**b**) SK90(thin film oven test (TFOT) specimens); (**c**) SK90 (pressure aging vesse (PAV) specimens); (**d**) Styrene-butadiene-styrene (SBS) (unaged specimens); (**e**) SBS modified asphalt (TFOT specimens); (**f**) SBS modified asphalt(PAV specimens); (**g**) SBS/crumb rubber powder (CRP) modified asphalt (Unaged specimens); (**h**) SBS/CRP modified asphalt (TFOT specimens); (**i**) SBS/CRP modified asphalt (PAV specimens).

**Figure 3 materials-13-04921-f003:**
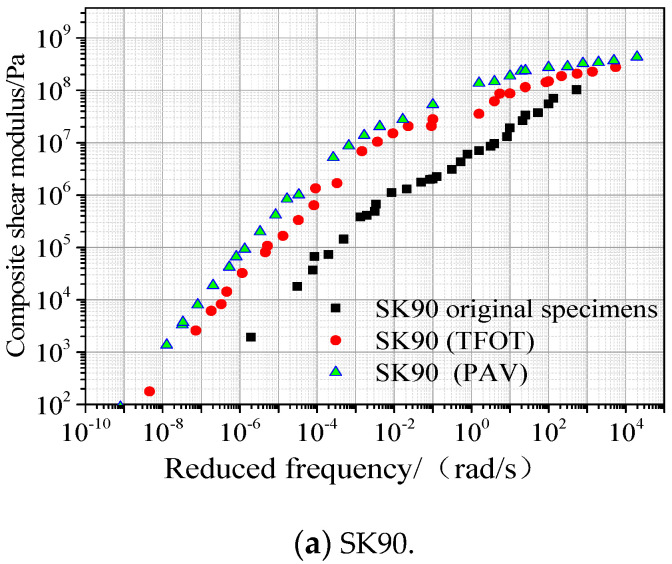
Master curve of composite shear modulus of asphalt: (**a**) SK90; (**b**) SBS; (**c**) SBS/CRP.

**Figure 4 materials-13-04921-f004:**
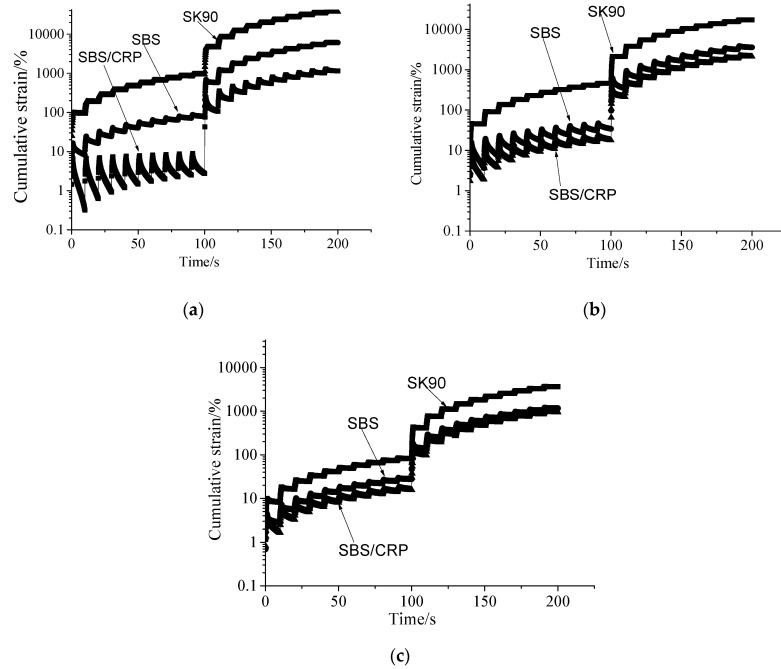
MSCR test curves of three asphalts at 64 ℃ under different aging conditions: (**a**) original asphalts; (**b**) asphalts after TFOT aging; (**c**) asphalts after PAV aging.

**Figure 5 materials-13-04921-f005:**
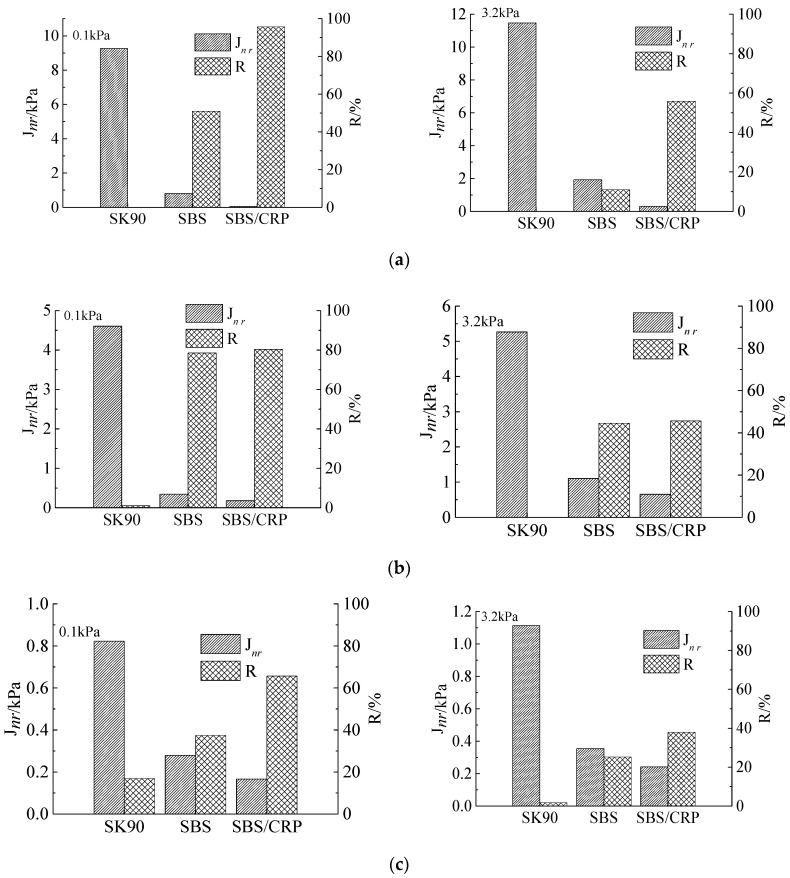
*J_n_*_r_, *R* at different aging conditions: (**a**) *J_nr_* and *R* of asphalt (unaged); (**b**) *J_nr_* and *R* (TFOT); (**c**) *J_nr_* and *R* (PAV).

**Figure 6 materials-13-04921-f006:**
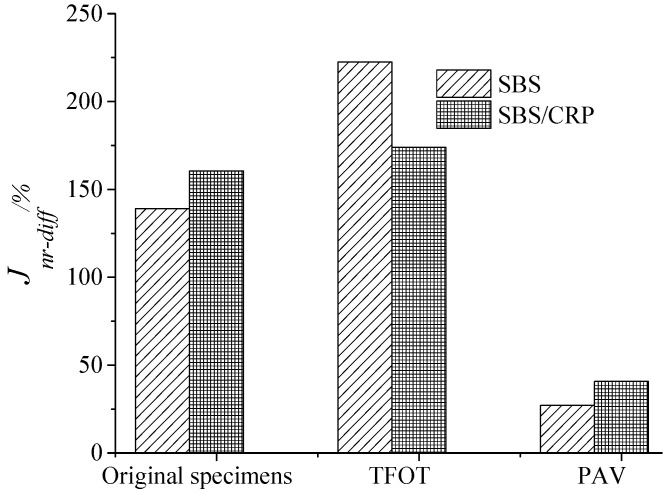
*J_nr-diff_* of three asphalts in different aging treatments.

**Table 1 materials-13-04921-t001:** Technical parameters of basic performances of SK90 asphalt.

Technical Indexes		Units	Test Results	Technical Requirements	Test Methods
Penetration (25 °C, 100 g, 5 s)		0.1 mm	94.6	80–100	T0604
Elongation (15 °C, 5 cm/min)		cm	>100	≮100	T0605
Softening Point (R & B)		°C	45.8	≮44	T0606
After TFOT	Mass loss	%	+0.4	±0.8	T0610
	Residual penetration ratio (25 °C)	%	57.8	≮57	T0604
	Residual ductility (10 °C)	cm	12	≮8	T0605

**Table 2 materials-13-04921-t002:** Basic performances of YH791.

Styrene to Butadiene	Density/(g/cm^3^)	Melt Flow Rate/(g/10 min)	Elongation at Break/%	Tensile Strength/MPa	Oil Filling Rate/%	Structure
30/70	0.94	0.50~5.00	58	12.0	0	linear

**Table 3 materials-13-04921-t003:** Technical indexes of crumb rubber powder [33,34,35,36].

Items		Units	Test Results	Technical Requirements	Test Methods
Physical indexes	Relative density	kg/cm^3^	1.15	≤1.2	GB/T 533-2008(Method B)
	Metal content	%	0.005	≤0.05	GB/T 19208-20206.9
Chemical indexes	Ash content	%	6.1	≤10	GB/T 4498-2013(Method B)
	Acetone-extracted content	%	7.1	≤10	GB/T 3516-2006(Method A)
	Carbon black content	%	30	≥26	GB/T 533-2008(Method B)
	Rubber hydrocarbon content	%	50.1	≥42	GB/T 1438-1993 7.1

**Table 4 materials-13-04921-t004:** Temperature shift factor.

Temperatures/°C	Original Specimens			TFOT			PAV		
	SK90	SBS	SBS/CRP	SK90	SBS	SBS/CRP	SK90	SBS	SBS/CRP
−20	2.22	1.67	1.39	1.74	2.25	1.65	0.73	1.60	1.71
−10	0	0	0	0	0	0	0	0	0
10	−3.68	−2.91	−2.51	−3.03	−3.78	−2.88	−1.50	−2.84	−2.91
30	−6.32	−5.27	−4.62	−5.48	−6.59	−5.19	−3.07	−5.23	−5.14
50	−7.94	−7.05	−6.35	−7.33	−8.41	−6.93	−4.70	−7.17	−6.70

**Table 5 materials-13-04921-t005:** Bending beam rheometer (BBR) test results.

Aging Degree	Asphalt Types	Values S and m at Different Temperatures							
		−12		−18		−24		−30	
		S	m	S	m	S	m	S	m
Original Specimens	SK90	119	0.376	276	0.288	583	0.218	-	-
	SBS	104	0.423	300	0.330	572	0.233	-	-
	SBS/CRP	81.7	0.445	186	0.359	270	0.308	398	0.264
TFOT	SK90	129	0.378	347	0.297	677	0.232	-	-
	SBS	102	0.411	253	0.323	516	0.240	-	-
	SBS/CRP	68.7	0.415	178	0.330	288	0.311	479	0.255
PAV	SK90	172	0.314	353	0.261	750	0.200	-	-
	SBS	165	0.371	296	0.307	533	0.231	-	-
	SBS/CRP	106	0.412	215	0.312	296	0.305	569	0.205

**Table 6 materials-13-04921-t006:** Parameters of Burgers model.

Parameters of Burgers Model		SK90	SBS	SBS/CRP
−12	*E*_1_/MPa	9000	8000	7000
	*E*_2_/MPa	617.3	342.6	266.1
	*η*_1_/MPa/s	40,971.9	52,584.8	23,164.1
	*η*_2_/MPa/s	7282.8	6781.8	4605.7
−18	*E*_1_/MPa	11,000	11,000	11,000
	*E*_2_/MPa	757.5	603.2	479.6
	*η*_1_/MPa/s	129,462	114,182	82,493.5
	*η*_2_/MPa/s	16,943.0	13,066.9	10,207.1
−24	*E*_1_/MPa	-	-	10,000
	*E*_2_/MPa	-	-	743.2
	*η*_1_/MPa/s	-	-	63,102.2
	*η*_2_/MPa/s	-	-	15,379.6

**Table 7 materials-13-04921-t007:** Parameters of comprehensive creep compliance.

	Asphalt Types	SK90	SBS	SBS/CRP
Indexes/(1/MPa)				
*J_E_*		1.11 × 10^−4^	1.25 × 10^−4^	1.43 × 10^−4^
*J_De_*		1.62 × 10^−3^	2.92 × 10^−3^	3.76 × 10^−3^
*J_V_*		3.30 × 10^−2^	3.54 × 10^−2^	5.21 × 10^−2^
*J*		3.13 × 10^−2^	3.26 × 10^−2^	4.85 × 10^−2^

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
