# Peer review of "Rheological Properties of SBS/CRP Composite Modified Asphalt under Different Aging Treatments"

_materials, 2020, doi:10.3390/ma13214921_

Round 1
Reviewer 1 Report
The comments have been addressed properly.
Author Response
There is no report of Reviewer 1.
Reviewer 2 Report
The authors have taken into consideration all the comments on the revised version.
Author Response
There is no report of Reviewer 2.
Reviewer 3 Report
In my opinion this paper should be comprehensively reviewed by authors.
The main issue is novelty of this work. The SBS/CR blends as modifiers of bitumen are known (from 2014 commercially availble in EU). The authors should highlight the novelty of presented work.
Moreover, all references were used in Introduction and test methods sections.
During description of results there is no comparison to results from other research group (this is not first work related to modificiation of bitumen by SBS/CR blends).
Please remove Figure 1 and 2 from manuscript, which showed the apparance of well known equipment.
More detailed comments in the attachment.

Reviewer 4 Report
Interesting topic and a well-written paper. Some questions:
1- Why you are using TFOT nor RTFOT for again?
2- I believe you do not need to show figure 4, figure 5 is enough.
3- Please improve the quality of figure 5, add some colours and show the best fit for the master-curve. The figure is not clear at all. The results are wired to me, why shear modulus is decreasing after ageing? is the labelling correct for figure 5?
4- Table 4 shows the low temperature for SBS/CPR is -24, why in the abstract and conclusion parts it has been mentioned as -34?
5- conclusion 4: low-temperature performance is -24, not -34, what is the required low-temperature performance of the alpine cold region? How much price is being added as a result of this modification?
6- Conclusion 4: How you are defining the construction (compaction) temperature? It is very important that you show a viscosity-temperature graph for this conclusion. Also, mixing temperature is important to be mentioned as well.
Round 2
Reviewer 3 Report
In my opinion Authors improved the manuscript. However, I still have some minor suggestions, which are summarized in attachment.

Reviewer 4 Report
I believe the authors have addressed al my comments. Thank you!
Author Response
There is no report. Thanks
This manuscript is a resubmission of an earlier submission. The following is a list of the peer review reports and author responses from that submission.
Round 1
Reviewer 1 Report
Please corect / complete the following:
- row 111: equal sign
- row 123: Jnr - subscript for nr
- row 253: text overlay for (b)
- row 260: Jnr - subscript for nr
- row 326: what is meant by "economic factors"? Details.

Reviewer 2 Report
What do you mean by using TFOT an PAV at the same time?
It is not clear what the novelty and innovation of this are. What are the new findings that your research adds to the current literature? The effects of SBS and CR as well as their coupled effects have been the focus of many previous studies. In the introduction section, the authors fail to cite the previous studies and to identify the gap in the current literature.
You have used 20 references just to support two short paragraphs. It would be suggested to reduce the number of references. You may want to use the following review paper to support paragraphs 1 and 2:
https://doi.org/10.1016/j.eurpolymj.2018.10.049
Please explain about the procedure for the selection of the percentage of the modifiers. Why 4.5% SBS? Why not 5.0%? or 4.0%? you have not mentioned about the dosage of CR?
Lines 88-89: Still, it would be suggested to add a brief description about the sample preparation.
The English of the paper needs a revision.
Why did you use BBR test for unaged and RTFOT-aged binders? BBR test is developed to study the low temperature properties where thermal cracking is the most common distress and aging can make it worse.
For your MSCR results, it seems that you have done 10 cycles at 0.1 kPa and 10 cycles at 3.2 kPa. The modified version of the MSCR tests should be done using 20 cycles at 0.1 kPa, where the first 10 cycles is used for conditioning the specimen. Please explain why you did only 10 cycles at 0.1 kPa?
MSCR test is developed to study the rutting potential of the binders. It is not clear why you conducted the tests on unmodified and PAV-aged binders. Please clarify.
Reviewer 3 Report
#1. Line 36-37: Please provide brief mechanisms or reasons for the statement with references.
#2. In section 2.1 raw materials, please provide the reason how the authors selected the dosages for SBS modifier, solubilizer, anti-aging agent, and auxiliary agent.
#3. Sections 2.2 – 2.5: Reviewer recommends adding some figures explaining testing procedures such as specimen photos, photos of the testing setup, and/or schematic testing procedures, etc.
#4. Line 151-177: It is very hard to read by comparing the text and figure 1. Please re-describe the explanation of the results in a different but more effective way. For example, you can use a table showing the changes of the modulus values expressing by percentage.
#5. In this paper, the authors experimented the effect of different additives for modifying rheological properties of asphalt mixture. However, although the test results are shown, there were barely discussed about possible mechanisms how the additives affect the properties. Thus, the reviewer suggests that the authors include scientific discussion about “why?” for the testing results throughout the manuscript.
#6. The authors concluded that the testing results support that the SBS/CRP composite modified asphalt can be used in most alpine cold regions due to low-temperature performance. However, the reviewer thinks this is the too early conclusion. This should be checked in a design perspective. How the better low-temperature performance of the composite influences on which design components, and how the design components affect end-performance of the asphalt mixture under the alpine region conditions.